# Lipidomics and Transcriptome Reveal the Effects of Feeding Systems on Fatty Acids in Yak’s Meat

**DOI:** 10.3390/foods11172582

**Published:** 2022-08-26

**Authors:** Lin Xiong, Jie Pei, Xingdong Wang, Shaoke Guo, Xian Guo, Ping Yan

**Affiliations:** 1Animal Science Department, Lanzhou Institute of Husbandry and Pharmaceutical Sciences, Chinese Academy of Agricultural Sciences, Lanzhou 730050, China; 2Key Laboratory of Animal Genetics and Breeding on Tibetan Plateau, Ministry of Agriculture and Rural Affairs, Lanzhou 730050, China; 3Key Laboratory for Yak Genetics, Breeding, and Reproduction Engineering of Gansu Province, Lanzhou 730050, China

**Keywords:** yak’s meat, candidate gene, feeding system, fatty acids, n-3PUFAs

## Abstract

The differences of fatty acids in yak’s meat under graze feeding (GF) and stall feeding (SF) regimes and the regulation mechanism of the feeding system on the fatty acids content in yak ’s meat was explored in this study. First, the fatty acids in yak’s longissimus dorsi (LD) muscle were detected by gas liquid chromatography (GLC). Compared with GF yaks, the absolute content of ΣSFAs, ΣMUFAs, ΣUFAs, ΣPUFAs and Σn-6PUFAs in SF yak’s LD were higher, whereas Σn-3PUFAs was lower; the relative content of ΣMUFAs, ΣPUFAs, Σn-3PUFAs and ΣUFAs in SF yak’s LD were lower, whereas ΣSFAs was higher. The GF yak’s meat is healthier for consumers. Further, the transcriptomic and lipidomics profiles in yak’s LD were detected by mRNA-Sequencing (mRNA-Seq) and ultra-high performance liquid chromatography-mass spectrometry (UHPLC-MS), respectively. The integrated transcriptomic and lipidomics analysis showed the differences in fatty acids were caused by the metabolism of fatty acids, amino acids, carbohydrates and phospholipids, and were mainly regulated by the *FASN*, *FABP3*, *PLIN1*, *SLC16A13*, *FASD6* and *SCD* genes in the PPAR signaling pathway. Moreover, the *SCD* gene was the candidate gene for the high content of ΣMUFA, and *FADS6* was the candidate gene for the high content of Σn-3PUFAs and the healthier ratio of Σn-6/Σn-3PUFAs in yak meat. This study provides a guidance to consumers in the choice of yak’s meat, and also established a theoretical basis for improving yak’s meat quality.

## 1. Introduction

With living standards improving, consumers are more and more focused on meat quality [1], and request to get more superior meat. Meanwhile, the meat industry focuses on developing valuable nutritional properties in meat. Moreover, meat safety incidents frequently happen in the world, and meat safety becomes a focus too [2]. The content and type of fatty acids in livestock meat are among the key factors which affect the sensory qualities of meat such as tenderness, color, and cooking loss [3,4]; on the other hand, the composition of fatty acids in livestock meat is a critical index in evaluating the meat nutrition. Saturated fatty acids (SFAs) in livestock meat can increase the risk of cardiovascular disease for consumers [5], while more polyunsaturated fatty acids (PUFAs), especially n-3 PUFAs like eicosapentaenoic acid (EPA) and docosahexaenoic acid (DHA), can protect blood vessels and prevent cardiovascular and cerebrovascular diseases [6,7]. The livestock meat with a lower ratio of Σn-6/Σn-3 PUFAs is a benefit to human health [8], so is very popular with consumers.

Researchers in livestock husbandry are attempting to alter the composition of fatty acids in livestock meat by increasing the content of unsaturated fatty acids (UFA), especially n-3 PUFAs in recent years. There are significant differences in digestion and absorption characteristics between monogastric and ruminant animals, and the rumen bacteria plays an important part in digestion and absorption for ruminant animals. In production practice, the proportion of concentrate feed for monogastric animals is usually higher, while the proportion of roughage for ruminants is higher. The rumen microbiome through the dietary ingredients under different feeding systems greatly affects the composition of fatty acids. Due to the hydrogenation of ruminal microorganisms in ruminants, the content of SFAs in ruminants’ muscle is usually higher than that of monogastric animals and the ratio of PUFA to SFA is lower [9,10]. The variation in nutrient composition in livestock meat is controlled by the genetics of the animal and the production environment of the livestock [11]. The effect of genetics, diet, gender and feeding system on the content and composition of fatty acids in livestock meat were proved [12,13].

The feeding system is one of the most important factors affecting the fatty acids in livestock meat [14]. Compared with livestock meat from stall feeding (SF), livestock meat from graze feeding (GF) tends to possess a higher percentage of n-3 PUFAs and conjugated linoleic acid [15]. The yak (*Bos grunniens*) is a unique animal species and one of the dominant animal breeds on the Tibetan Plateau [16]. Nowadays, the yield of yak’s meat in the world exceeds 300 thousand tons per year. Because of high protein content, plenty of functional fatty acids, a unique flavor and an absence of pollutants, yak’s meat has widely attracted the attention of the market [17,18]. The content of SFAs in yak muscle was lower than the values in cattle, while the contents of PUFAs and monounsaturated fatty acids (MUFAs) in yak muscle were higher than the values in cattle [19,20]. The ratio of n-6 PUFAs/n-3 PUFAs in muscle of yak calf was 1.15 [21]. The traditional grazing system is still predominant in yak industry [22], so the yak’s growth is greatly affected by the natural conditions including temperature, precipitation, and grass status. The emergence rate and production performance of yaks are lower by contrast with other cattle breeds. The optimum slaughter ages for yaks is 3.5–5 years longer than 1.5–2 years for beef cattle [23]. Moreover, grass in the Tibetan Plateau has a very short growth period due to the cold weather, and overgrazing has led to the serious grassland degradation at present [24], which seriously threatens the fragile ecology of Tibetan Plateau. In recent years, the SF is being applied in yak production to improve production efficiency and protect the environment [25]. The change of feeding system for yaks maybe lead to differences in the fatty acids in yak’s meat. But as far as we know, there are no reports on the effect of the feeding system on the fatty acids in yak’s meat and the reasons for the differences.

The content of fatty acids in meat is the result of the metabolism of fatty acids in livestock muscle. Compared with monogastric animals, the fat metabolism in ruminants is more complex. The rumen microorganisms in ruminants can translate fatty acids from forage or feed into new fatty acids [26], and finally these new fatty acids deposit in subcutaneous, muscle and the abdomen. It is difficult to explore the effect of the feeding system on fatty acids in yak’s meat using a single molecular biology technique. Existing research suggests that the feeding system adjusts the content of fatty acids in livestock meat by rescheduling the metabolite concentration and the expression level of genes involving in fatty acids metabolism (like *LIPC*, *ERFE*, *FABP3*, *PLA2R1*, *LDLR*, and *SLC10A6* and so on) [27]. Transcriptomics can study the transcription and regulation of all genes in cells at the global level and has been applied to the studies of fatty acids in livestock like cattle [28,29], pig [30], goat [31] and yak [32]. The newest RNA-sequencing (RNA-seq) analysis can examine the mRNA level in yak muscle under different feeding system and estimate gene expression profile. Lipidomics is a kind of high-throughput analysis technology and can systematically analyze the changes of lipid composition and expression in organisms [33] and explore the functions of lipid families and lipid molecules in various biological processes [34]. There are some reports on the studies of the characterization of fatty acids in dairy [35,36], cattle [37,38], goat [39] and pork [40,41] by lipidomics. The lipidomics of yak muscle can help to identify the metabolic networks of fatty acids, which can affect the content and composition of fatty acids. The fatty acids phenotype-related gene functions and lipid metabolism pathways in yak muscle under different feeding system can be revealed by the approach combined lipidomics and transcriptomics. 

In this study, we hypothesized that the feeding system could affect the fatty acids metabolism in yak muscle by the differential expression of genes associated with fatty acids, and ultimately cause a change to the content and composition of fatty acids in it. The absolute and relative content of fatty acids in yak’s longissimus dorsi (LD) muscle were determined, and the differences in the characteristics of fatty acids in yak muscle between GF and SF were observed. Further, the differentially expressed genes (DEGs) and significantly different lipids (SDLs) in GF and SF yak’s LD were identified by RNA-Seq transcriptomics and non-targeted lipidomics, respectively. The DEGs and SDLs were annotated by Gene Ontology (GO) and Kyoto Encyclopedia of Genes and Genome (KEGG), then the molecular mechanism of the feeding system affecting fatty acids in yak muscle was explored. This study investigated the differences of fatty acids in yak’s LD under SF and GF and established a theoretical basis for the evaluation of yak’s meat under different feeding system and provided a reference for improving intramuscular fat (IMF) content in yaks.

## 2. Materials and Methods

### 2.1. Animals and Sample Collection

The animal experiment was approved by the Ethics Committee of the Lanzhou Institute of Husbandry and Pharmaceutical Sciences, Chinese Academy of Agricultural Sciences (Permit No. SYXK-2020-0166). A total of twelve healthy male yaks (the same genetic background, two-year old, 210.33 ± 10.23 kg) were selected in May and were divided into two groups in the study. The GF group: six yaks (*n* = 6) were only grazed in natural pasture with no supplements; the SF group: six yaks (*n* = 6) were fed with total mixed ration (TMR) food in a stall. The GF and SF yaks were fed in Qinghai province in China, and the yaks in each group can freely eat grass or MTR and drink water by themselves, respectively. All experimental animals were dewormed before the test. By September, all yaks were humanely slaughtered by professional technicians at a commercial abattoir. The slaughter procedure was conducted in accordance with European Commission Regulation. The LD samples from between the 12th and 13th ribs of the left side of the carcass were immediately collected after the yaks were slaughtered. Part of the LD samples were put in liquid nitrogen immediately, and the rest were frozen at −20 °C. The grass sample was collected in September too. The composition of TMR and the content of common nutrition and major fatty acids in grass and TMR are shown in Table 1.

### 2.2. Determination of Fatty Acids

The absolute content of fatty acids in the yak’s LD was detected according to the method described in references [42,43,44] with some modifications. First, the lipid in yak’s LD was extracted with a mixed solvent of chloroform and methanol (*v:v*, 2:1) by Soxhlet extraction for three times, then the combined extract solution was dried under blown nitrogen in a water bath. Second, the extracted lipid in the yak’s LD was decomposed into free fatty acids with a sodium hydroxide methanol solution by a basic hydrolysis in a water bath. Third, the fatty acids were transferred into the fatty acid methyl esters (FAMEs) by an esterification reaction with boron fluoride-methanol solution. The FAMEs were extracted with normal heptane and were dried under nitrogen. The residue was redissolved with a mobile phase, then was determined on an gas chromatography system (7890A, Agilent Corp., Santa Clara, CA, USA). The analytes were confirmed by comparing their retention times with the values of standard compounds. The absolute content of fatty acids was calculated by the absolute content of FAMEs, and the relative content was calculated as follows: the relative content of fatty acid = (the absolute content of a certain fatty acid)/(the absolute content of total fatty acids) × 100.

### 2.3. RNA Extraction and Sequencing

Three samples of yak’s LD were randomly selected from six samples in the SF and GF groups, respectively. According to the protocol of the manufacturer, the total RNA in yak’s LD was extracted with the mirVanaTM miRNA Isolation Kit (Ambion Inc., Foster City, CA, USA). The purity of the extracted RNA was determined with a NanoDrop 2000 spectrophotometer (Thermo Fisher Scientific, Santa Clara, CA, USA). Moreover, the RNA integrity was evaluated with the Agilent 2100 Bioanalyzer (Agilent Technologies, Santa Clara, CA, USA). The mRNA libraries for sequencing were prepared using TruSeq Stranded mRNA LTSample Prep Kit (Illumina, San Diego, CA, USA), then these libraries were sequenced on the Illumina sequencing platform (HiSeqTM 2500) and 125 bp paired-end reads were generated. Raw data containing ploy-N and the low-quality reads were removed, then the clean reads were obtained. The clean reads were mapped to a reference genome using hisat2 and the Q30 and GC-content of the clean reads were calculated. The numbers of reads were normalized against the reads per kilo base of transcripts per million (RPKM) to compute the gene expression levels. The RPKM value of each gene was calculated using cufflinks, and the read counts of each gene were obtained by htseq-count.

### 2.4. Lipids Extraction and MS Data

Fifty mg of the LD sample was put in an eppendorf (EP) tube, and Lyso PC17:0 solution in methanol, internal standard 2-chloro-l-phenylalanine solution in methanol and pure water were successively added to the tube. The LD sample was broken by ultrasonication, then the mixture of solutions and LD sample was transferred into a centrifuge tube. The solution of chloroform-methanol was added into the centrifuge tube, and the mixture was vortexed for 1 min. After centrifuging, the subnatant was transferred into a sample bottle and dried. The residue was dissolved in the solution of isopropanol-methanol and vortexed for 30 s. The solution was transferred into an EP tube and centrifuged. Two hundred μL supernatant was collected and filtered into a LC vial. A Nexera UPLC (Shimadzu, Kyoto, Japan) with Waters ACQUITY UPLC BEH C_18_ (100 mm × 2.1 mm, 1.7 μm) was used to separate the extract. The moving phase was composed of (A) acetonitrile and water (*v:v*, 60:40), containing 0.1% formic acid and 10 mmol/L ammonium formate and (B) acetonitrile and isopropyl alcohol (*v:v*, 10:90), containing 0.1% formic acid and 10 mmol/L ammonium formate. The column temperature, the flow rate of moving phase and injection volume were 45 °C, 0.3 mL/min and 5 μL, respectively. The Q Exactive MS system (Thermo Scientific™, Waltham, MA, USA) was used to collect the MS data and the parameters of the MS system were as follows: Positive ion mode—heater temp: 300 °C; the flow rate of sheath gas: 45 arb; the flow rate of aux gas: 15 arb; the flow rate of sweep gas: 1 arb; spray voltage: 3.5 KV; capillary temp: 320 °C; S-Lens RF level: 50%; MS1 scan ranges: 120–1800. Negative ion mode—heater temp 300 °C; the flow rate of heath gas: 45 arb; the flow rate of aux gas: 15 arb; the flow rate of sweep gas: 1 arb; spray voltage: 3.1 KV; capillary temp: 320 °C; S-Lens RF level 50%; MS1 scan ranges: 120–1800.

### 2.5. Quantitative Real-Time PCR (qPCR) 

Six DEGs, including *PLIN2*, *FABP3*, *LEP*, *SCD*, *ACAA1* and *ACOT7*, were selected to confirm the mRNA-Seq results. The information of primers is listed in Appendix A. The reference gene for qPCR in the yak muscle was the *β-actin* gene. The qPCR analysis for the target genes was carried out with ABI Prism 7500 instrument (Applied Biosystems, Carlsbad, CA, USA). A volume of 1.5 µg RNA, 0.5 µL gDNA remover, 5 µL 5 × TransScript All-in-one SuperMix, 0.5 µL of each primer and nuclease-free water were used in reverse transcription (RT) reaction. The reactions were carried out on a GeneAmp^®^ PCR System 9700 thermal cycler (Applied Biosystems, Foster City, CA, USA) for 15 min at 42 °C and 5 s at 85 °C. Next, the mixture for the RT reaction was diluted 10 times with nuclease-free water, then stored at 20 °C. The Real-time PCR was carried out on a LightCycler^®^ 480 Real-time PCR Instrument (Roche, Basel, Switzerland), and 0.2 µL cDNA, 5 µL 2 × PerfectStartTM Green qPCR SuperMix, 0.2 µL of each primer and 3.6 µL nuclease-free water were used in a 10 µL PCR. The parameters for the Real-time PCR were 94 °C for 30 s, followed by 45 cycles of 94 °C for 5 s and 60 °C for 30 s. Relative gene expression levels were determined using the 2^−∆∆Ct^ method [45].

### 2.6. Statistical Analysis

The data of the fatty acids was analyzed with the independent-sample T test in the software SPSS 16.0 (SPSS Inc., Chicago, IL, USA), and the results are shown by means ± standard error of the mean (SEM). Correlations among the fatty acids, DEGs and SDLs in yak’s LD were conducted by Pearson correlation analysis in SPSS 16.0 too. Significant differences and correlation were considered at *p* < 0.05 and correlation coefficient > 0.8 or <−0.8. These genes with fold change > 2 or < 0.5 and *p* < 0.05 were deemed as DEGs. The original Q Exactive mass data in raw format were processed with the software Lipid Search. A data matrix was combined from the positive and negative ion data. The matrix was imported into R to carry out principal component analysis (PCA). Further, orthogonal partial least squares discriminant analysis (OPLS-DA) was used to distinguish the lipids which were different in yak LD between the GF and SF groups. The SDLs were selected with VIP > 1.0 and *p* < 0.05. The GO analysis for DEGs was carried out with R based by running queries for each DEGs against the GO database. The KEGG analysis for DEGs and SDLs was performed to identify those KEGG pathways with *p* < 0.05.

## 3. Results

### 3.1. Content of Fatty Acids

The results of absolute and relative content of fatty acids in GF and SF yak’s LD are shown in Table 2. The absolute contents of C4:0, C6:0, C10:0, C11:0, C12:0, C13:0, C14:0, C16:0, C18:0, C21:0, C22:0, C24:0, *c*-C18:1, *t*-C18:1, *t*11-C18:1, *c*-C20:1, C24:1, *c*-C18:2n6, c9, *t*11-C18:2, *t*11, *c*15-C18:2, C20:4n6, ΣSFAs, ΣUFAs, ΣMUFAs, ΣPUFAs, Σn-6 PUFAs and ΣBIs in SF yak’s LD were higher than these values in GF yak’s LD (*p* < 0.05), whereas the absolute contents of C8:0, C15:0ai, C14:1, *c*-C15:1, *c*-C17:1, *c*-C18:3n6, C18:3n3, *c*-C20:2, *c*-C20:5n3 and Σn-3 PUFAs in SF yak’s LD were lower than these values in GF yak’s LD (*p* < 0.05). Meanwhile, the value of Σn-6/Σn-3 PUFAs in SF yak’s LD was higher than the value in GF yak’s LD (*p* < 0.01), whereas the value of ΣPUFAs/ΣSFAs in SF yak’s LD was lower than the value in GF yak’s LD (*p* < 0.01). On the other hand, the relative contents of C16:0, C18:0, *c*-C17:1, *c*-C20:1, *c*-C18:2n6 and ΣSFAs in SF yak’s LD were higher than these values in GF yak’s LD (*p* < 0.05), whereas the relative contents of C8:0, C15:0, C15:0iso, C15:0ai, C16:0iso, C17:0, C17:0iso, C17:0ai, C14:1, *c*-C15:1, C16:1, *c*-C17:1, *t*11-C18:1, *c*9, *t*11-C18:2, *t*11, *c*15-C18:2, *c*-C18:3n6, C18:3n3, *c*9, *t*11, *c*15-C18:3, *c*-C20:2, *c*-C20:5n3, *c*-C22:6n3, ΣMUFAs, ΣPUFAs, ΣUFAs, Σn-3PUFAs and ΣBIs in SF yak’s LD were lower than these values in GF yak’s LD (*p* < 0.05).

### 3.2. Lipidomics Results

The PCA score plots for the samples of GF yak’s LD, SF yak’s LD and quality control (QC) is shown in Figure 1a. It can be found that the QC group samples were congregated tightly in a small area, which indicated that the lipidomics platform UHPLC-MS possessed high stability and the analysis possessed excellent reliability. Further, the OPLS-DA analysis showed the variables responsible for differentiation in yak’s LD between GF and SF (Figure 1b). As observed herein, R2Y(cum) = (0, 0.638) and Q2(cum) = (0, −0.734) indicated the model was robust and without overfitting (Figure 1c). It was found that there was an obvious difference between GF and SF yak’s LD, which demonstrated the feeding system can significantly affect the lipids in yak’s LD. Lipids with significant differences between the GF and the SF group are shown in volcano plots (Figure 1d). A total of 75 SDLs in yak’s LD under GF and SF were obtained (Appendix A). Of these, 26 SDLs were up-regulated in LD of SF yaks, whereas 49 SDLs were down-regulated (Figure 2a). A total of 11 SDLs were enriched in 17 KEGG pathways (Appendix A), and the bubble chart of KEGG pathways is shown in Figure 2b. The KEGG pathway being relevant to fat acids metabolism included glycerophospholipid metabolism (bom00564), fat digestion and absorption (bom04975), regulation of lipolysis in adipocytes (bom04923), insulin resistance (bom04931), linoleic acid metabolism (bom00591), alpha-linolenic acid metabolism (bom00592) and arachidonic acid metabolism (bom00590). The above KEGG pathways mainly focus on UFAs metabolism and glycerophospholipid metabolism. The crucial SDLs included PC(16:0/20:4), PS(18:0/18:1), TG(16:0/16:1/18:1), TG(16:0/18:1/18:1), TG(16:1/16:1/18:1), TG(16:1/18:1/18-:1), TG(16:1/18:1 /18:2), TG(18:0/18:1/18:1) and TG(18:1/18:1/18:2), and were derived from C16:0, C16:1, C18:0, C18:1, C18:2 and C20:4.

### 3.3. Transcriptome Results

The results of the PCA displaying the distinct biological variation among GF and SF yak’s LD are shown in Figure 3a, and the effect of feeding system to the yak’s muscle was obvious. The DEGs in LD of GF and SF yaks are shown in Appendix A. Compared with LD of GF yaks, there were 1682 DEGs in LD of SF yaks, and 429 DEGs were up-regulated, while 1253 DEGs were down-regulated. The heatmap of the DEGs in GF and SF yak’s LD is shown in Figure 3b. After GO enrichment of DEGs, the biological processes involving to fatty acids contained fatty acids metabolism process, glucose metabolic process, regulation of protein processing (Figure 4a). The DEGs were enriched in 45 major KEGG pathways (Appendix A), and the KEGG pathways involving to fatty acids included PPAR signaling pathway (bom03320), glycolysis/gluconeogenesis (bom00010), glycine, serine and threonine metabolism (bom00260), carbohydrate digestion and absorption (bom04973), fatty acid biosynthesis (bom00061), protein digestion and absorption (bom04974) (Figure 4b). The enrichment scores of PPAR signaling pathway (3.70) and fatty acid biosynthesis (3.52) were the highest in all KEGG pathways, and contained 19 DEGs and 4 DEGs, respectively. The heatmap of the crucial DEGs (*PLIN2*, *DBI*, *CPT2*, *FASN*, *FABP3*, *FADS6*, *SLC16A13*, *SCD*, *LEP*, *ACOT7*, *ACAA1* and *ACSL6*) involving the regulation of fatty acids content is shown in Figure 4c. 

### 3.4. Verification of mRNA Sequencing Using qPCR

As shown in Figure 4d, by contrast with GF yaks, the expressions of *PLIN2*, *FABP3*, *LEP* and *SCD* genes in the LD of SF yaks were up-regulated, whereas the expressions of *ACAA1* and *ACOT7* genes were down-regulated. All six DEGs in the LD of SF and GF yaks possessed the similar expression patterns in qPCR and mRNA-Seq data, which indicated the reliability of mRNA-Seq data for yak’s LD in this study.

### 3.5. Joint Analysis of the Lipidomics and Transcriptome

The joint analysis of the transcriptome and lipidomics data for yak’s LD under SF and GF was carried out to explore the molecular mechanism for the differences of lipid metabolism in LD of SF and GF yaks. The regulatory processes of crucial DEGs to the lipid metabolism was shown in Figure 5. The biggest differences in lipid metabolism between SF and GF yak’s LD relates to the phospholipid metabolism and triglyceride metabolism by the regulation of DEGs in PPAR signaling pathway.

### 3.6. Correlation of Fatty Acids, DEGs and SDLs

Results of Pearson correlation analysis among the fatty acids, crucial DEGs in fatty acids regulation, crucial SDLs in fatty acids metabolism are shown in Figure 6. The absolute content of ∑MUFAs and ΣUFAs were significantly positively correlated with the expressions of the *FASN*, *FABP3*, *PLIN2*, *SLC16A13*, *DBI* and *SCD* genes, and the concentrations of TG(16:0/16:1/18:1), TG(16:0/18:1/18:1), TG(16:1/16:1/18:1), TG(16:1/18:1-/18:1), TG(16:1/18:1/18:2) and TG(18:0/18:1/18:1); the absolute content of ∑SFAs and Σn-6PUFAs were significantly positively correlated with the expressions of the *FASN*, *FABP3*, *PLIN2* and *DBI* genes; the absolute content of Σn-3PUFAs was significantly positively correlated with the expression of the *FADS6* gene, whereas it was significantly negatively correlated with the expressions of the *FABP3*, *PLIN2*, *SLC16A13* and *DBI* genes; Σn-6/Σn-3 PUFAs showed significantly positive correlation with the expressions of the *FABP*3, *PLIN2*, *CPT2*, *SLC16A13* and *DBI* genes, whereas there was significantly negative correlation with the expression of the *FADS6* gene; ΣPUFAs/ΣSFAs showed significantly positive correlation with the expression of the *ACSL6* gene, whereas they were significantly negatively correlated with the expressions of the *FASN*, *FABP3*, *PLIN2*, *CPT2*, *SLC16A13*, *DBI* genes, and the concentrations of TG(16:0/16:1/18:1), TG(16:0/18:1/18:1), TG(16:1/16:1/18:1) and TG(16:1/18:1/18:1). 

## 4. Discussion

The deposition velocity of SFAs and MUFAs is faster than the value of PUFAs, which results in the decrease in relative content of PUFAs and the ratio of PUFA/SFA [46]. Compared with the grass-fed beef, the beef fed on TMR possesses higher SFAs and n-6 PUFAs, and lower n-3 PUFA [47]. The content of fatty acids in yak’s LD under different feeding systems showed the SF can also decrease the relative content of Σn-3 PUFAs, ΣPUFAs, and the ratio of ΣPUFAs/ΣSFAs, whereas it increases the relative content of ΣSFAs. The World Health Organization and Food and Agriculture Organization of United Nations made a suggestion that the intake of SFAs and trans fatty acids (TFAs) should be decreased, while the intake of n-3 PUFAs should be increased. The ratio of ΣPUFAs/ΣSFAs in meat should be more than 0.4, while the ratio of Σn-6/Σn-3 PUFAs should be less than 4 [48,49]. Three TFAs (t-C18:1, t11-C18:1 and t-C18:2n6) were simultaneously determined in LD of yaks under GF and SF, and the absolutely content of *t*-C18:1 and *t*11-C18:1 in LD of SF yaks were higher than these values in the GF group. The ratio of PUFAs/SFAs and n-6/n-3 PUFAs in GF yak’s LD were more than 0.4 and less than 4, respectively; whereas these values in LD of SF yaks were less than 0.4 and more than 4, respectively. Therefore, the meat of GF yaks is healthier than the meat of SF yaks from the viewpoint of nutritional value. Fat metabolism in ruminants is closely related to rumen microorganisms and the feeding system directly affects the rumen microorganisms. The biohydrogenation by rumen microorganisms in ruminants is the important way of UFAs transferring into MUFAs or SFAs [50]. The differences of rumen microorganisms in yaks under different feeding systems can cause the differences of biohydrogenation for UFAs in the diet, which is one of major factors that led to the differences in the fatty acids composition in yak muscle. 

The lipidomics showed the fatty acids in yak’s LD mainly existed in the form of triglyceride and phosphatidylcholine, and the major fatty acids in SDLs were C16:0, C16:1, C18:0, C18:1 and C18:2. Moreover, the determination of the fatty acids revealed the major ones in yak’s LD under both SF and GF were C16:0, C16:1, C18:0, C18:1 and C18:2, and the absolute contents of C16:0, C18:0, C18:1 and C18:2 in GF and SF yak’s LD were different. So the differences of fatty acids in SF and GF yak’s LD were mainly due to C16:0, C18:0, C18:1 and C18:2. The DEGs in GY and SY yak’s LD were mainly involved in the regulation of the fatty acids metabolism including fatty acids biosynthesis (bom00061), fat digestion and absorption (bom04975), amino acid metabolism including glycine, serine and threonine metabolism, protein digestion and absorption (bom04974) and carbohydrate metabolism including glycolysis/gluconeogenesis (bom00010), and carbohydrate digestion and absorption (bom04973). The fatty acids, amino acids and carbohydrate could transform to each other by the tricarboxylic acid (TCA) cycle [51,52]. Therefore, it can be inferred that the effect of the feeding system on the fatty acids in yak muscle was realized by the reciprocal transformation of three major macronutrients. The carbohydrate and amino acid intake by yaks from the grass or forage can transfer into short chain fatty acids, then the short chain fatty acids were further transformed to long chain SFAs. Moreover, the long chain SFAs can translate into MUFAs. In a word, the GF and SF yaks can intake different carbohydrates, amino acids, and fatty acids from their diet, which lead to the differences in fatty acids in yak muscle in the larger sense. 

The regulation of fatty acid transport, fatty acid catabolism and fatty acid anabolism in bovines are driven by these genes in the PPAR signaling pathway [53]. The transcriptome analysis showed the differences of fatty acids in GS and SF yaks were mainly realized by the regulation of the PPAR signaling pathway too, and these downstream genes (*FABP*, *FASN*, *ME1*, *SCD*, *ACBP*, *LPL*, *CPT1*, *ACSL*s, *ACAA1* and *PLINs*, which are also involved in the regulation of fatty acid metabolism in beef cattle [54,55]) in the PPAR signaling pathway were essential in the regulation of fatty acids metabolism in yak muscle. On the other hand, the SDLs were mainly enriched in phospholipid metabolism like glycerophospholipid metabolism (bom00564), glycerolipid metabolism (bom00564), triglyceride metabolism like fat digestion and absorption (bom04975), and UFAs metabolism like linoleic acid metabolism (bom00591), alpha-linolenic acid metabolism (bom00592), arachidonic acid metabolism (bom00590). Therefore, the fatty acids in yak muscle under different feeding systems were directly affected by the metabolism of UFAs, phospholipids and triglycerides regulated by the related genes in the PPAR signaling pathway.

As can be seen from the angle of molecular biology, the content of fatty acids in yak muscle is largely determined by some crucial enzymes in fatty acids metabolism. Identification of the genes encoding these enzymes can help understand the genetic variation underlying the content of fatty acids in yak meat. The *PLIN1* gene negatively regulates the lipolysis in dairy cows [56]. The expression of the *PLIN1* gene in yak muscle was positively correlated with the absolute content of ∑SFAs, ∑MUFA, Σn-6PUFAs, ΣUFAs and the ratio of Σn-6/Σn-3 PUFAs, so the *PLIN1* gene in yak muscle can positively regulate all types of fatty acids content except Σn-3PUFAs, which is due to preventing fatty acid degradation. Biosynthesis of SFAs can be realized by the de novo synthesis of fatty acids and elongation of fatty acids, and biosynthesis of MUFAs can be realized by elongation of fatty acids and dehydrogenation of SFAs. The expressions of *FASN*, *FABP3*, *SLC16A13* and *DBI* genes in yak muscle were positively correlated with the absolute content of ∑SFAs and ∑MUFAs, respectively. Moreover, the expression of the *SCD* gene in yak muscle was positively correlated with the absolute content of ∑MUFAs. The expression of the *FASN* gene was significantly associated with the concentrations of C14:0, C16:0 and C18:1 in cattle muscle [57]; the *FABP3* plays an important role in the oxidation, esterification and metabolism of SFAs and MUFAs [58]; the *DBI* gene regulates the synthesis and degradation of ketone bodies, which is a key step in glucose and amino acid translating to SFAs [59]; the *SCD* gene is responsible for the dehydrogenation in the biosynthesis of MUFAs especially *c*-C18:1 in Holstein cattle [60]; the *SLC16A13* gene positively regulates the long chain fatty acids transport which is necessary for the biosynthesis of long chain SFAs and MUFAs [61]. The absolute content of *c*-C18:1, ∑SFA and ∑MUFAs in SF yak’s muscle were higher than values in GF yak’s muscle. Therefore, *FASN*, *FABP3*, *SLC16A13* and *DBI* genes in yak’s muscle positively regulate the biosynthesis of SFAs and MUFAs by de novo synthesis of fatty acids, elongation of fatty acids, whereas the *SCD* gene in yak muscle was the only controlling gene in the process of SFAs translating to MUFAs. Some complicated PUFAs cannot directly be synthesized in liver and adipose tissue of livestock and must be transferred from the precursor compounds PUFAs from grass and feed [62]. The n-6 PUFAs like arachidonic acid are derived from linoleic acid by the action of desaturases and elongases, and n-3 PUFAs like DHA and EPA are derived from linolenic acid. The *FADS6* gene acts in the dehydrogenation of linoleic acid and linolenic acid, and plays a crucial role in PUFA biosynthesis [63,64]. The concentrations of C18:3n3, *c*-C20:5n3 and Σn-3PUFAs in SF yak muscle were lower than the values in GF yak muscle. Meanwhile, the expression of the *FADS6* gene in yak muscle was positively correlated with the absolute content of Σn-3PUFAs, and negatively correlated with the value of Σn-6/Σn-3 PUFAs. Therefore, the *FADS6* gene is a crucial candidate gene for regulating the Σn-3 PUFAs content and the ratio of Σn-6/Σn-3 PUFAs in yak muscle.

## 5. Conclusions

The characteristics and regulation of fatty acids in yak muscle under GF and SF were investigated in this study. The absolute content of ΣSFAs, ΣUFAs, ΣPUFAs, ΣMUFAs and Σn-6 MUFAs in SF yak’s meat were higher than these values in GF yak’s meat, whereas the relative content of Σn-3PUFAs, ΣPUFAs, ΣMUFA and ΣUFAs in SF yak’s meat were lower. The ratio of Σn-6/Σn-3 PUFAs, ΣPUFA/ΣSFA in GF yak’s meat was less than 4, and more than 0.4, respectively, thus the composition of fatty acids in GF yak’s meat is healthier for consumers from the point of nutritional value. The metabolism of fatty acids, amino acids, carbohydrates and phospholipids in GF and SG yak’s meat were different, and the differences of fatty acids in yak meat were mainly regulated by the *FASN*, *FABP3*, *PLIN1*, *SLC16A13*, *FASD6*, *DBI* and *SCD* genes in the PPAR signaling pathway.

## Figures and Tables

**Figure 1 foods-11-02582-f001:**
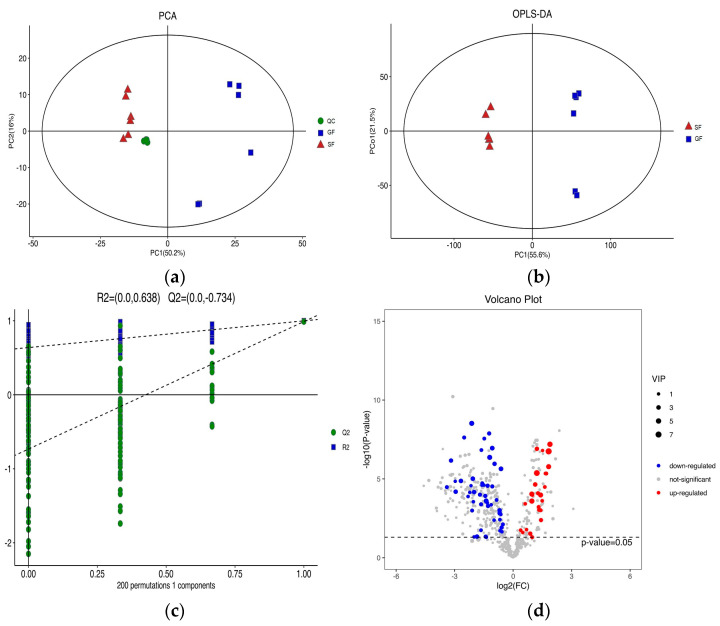
(**a**) The score plots of principle component analysis (PCA) for the lipids in yak’s longissimus dorsi (LD) muscle under stall feeding (SF) and graze feeding (GF) regimes. Triangle, square and circle represent SF yaks, GF yaks and quality control, respectively. (**b**) The result of orthogonal partial least squares discrimination analysis (OPLS-DA) for lipids in yak’s LD under SF and GF. (**c**) Scores of the overall sample and permutations test of OPLS-DA. (**d**) The volcano plot of total lipids in LD of SF yaks by contrast with GF yaks. Red dots represent the significant differences lipids (SDLs) whose concentration was up-regulated in SF yak LD, while the blue dots represent the SDLs whose concentration was down-regulated and grey dots represent non-SDLs.

**Figure 2 foods-11-02582-f002:**
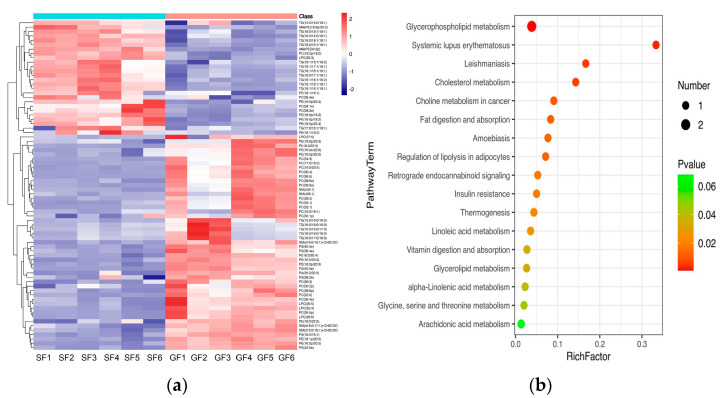
(**a**) The heatmap of SDLs in the yak’s LD under SF and GF. In the color scale, red and blue indicate the increase and decrease of SDL’s concentration, respectively. (**b**) Bubble chart of Kyoto Encyclopedia of Genes and Genomes (KEGG) enrichment analysis for SDLs in yak’s LD under SF and GF. The horizontal axis and the vertical axis represent the rich factor and the pathway term, respectively. Rich Factor is the ratio of the number of SDLs annotated in a certain KEGG pathway item with the number of all lipids annotated in this KEGG pathway item. The size and color of the bubble represent the number of DEG enriched in the KEGG pathway and the different *p*-value range, respectively.

**Figure 3 foods-11-02582-f003:**
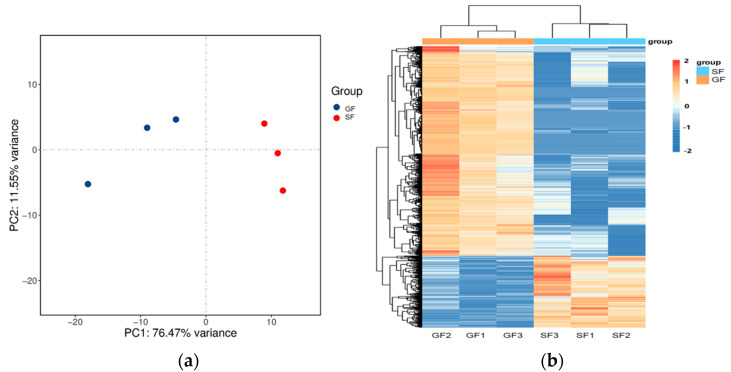
(**a**) The result of PCA analysis for the samples of yak’s LD under GF and SF. A circle represents a LD sample. Red is the SF yak’s LD, and blue is the GF yak’s LD. (**b**) The heatmap of the differentially expressed genes (DEGs) in GF and SF yak’s LD. The columns, rows and color scale represent individual samples, each DEG and the relative expression level of the DEGs, respectively.

**Figure 4 foods-11-02582-f004:**
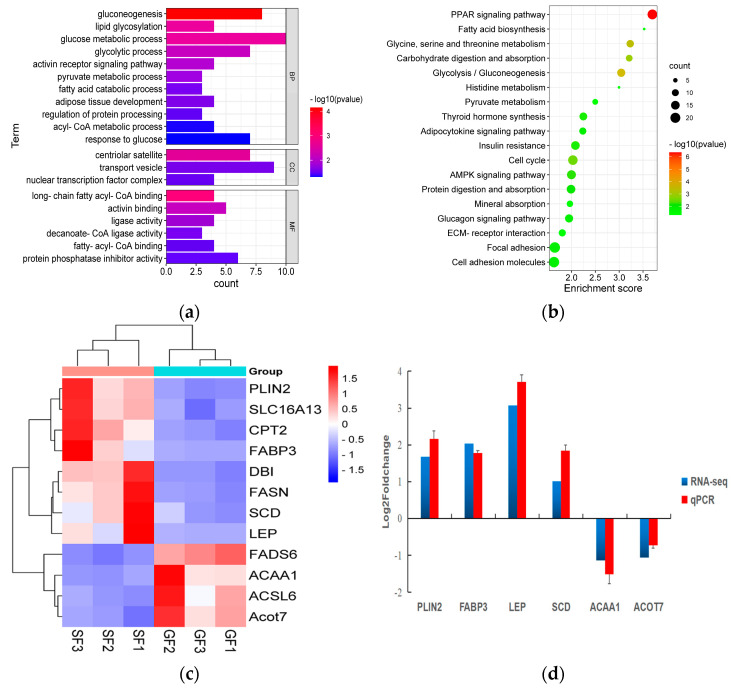
(**a**) The histogram of the Gene Ontology (GO) enrichment analysis of differentially expressed genes (DEGs) in yak LD under GF and SF. The items of GO enrichment analysis are divided into biological processes (BPs), cellular components (CCs) and molecular functions (MFs). (**b**) The bubble chart of the KEGG pathway enrichment analysis of DEGs in yak’s LD under GF and SF. The abscissa represents the enrichment score which is the ratio of the number of DEGs in a certain KEGG item with the total number of backgrounds identified in the KEGG pathway. The color represents the value of -log10 *P*, and the size of circle represents the number of DEGs in the KEGG item. (**c**) The heatmap of the crucial DEGs (*PLIN2*, *SLC16A13*, *CPT2*, *FABP3*, *DBI*, *FASN*, *SCD*, *LEP*, *FADS6*, *ACAA1*, *ACSL6* and *ACOT7*) involving in the regulation of fatty acids content in yak’s LD under SF and GF. (**d**) The comparison for the values of log_2_ fold change (FC) in six DEGs (*PLIN2*, *FABP3*, *LEP*, *SCD*, *ACAA1* and *ACOT7*) between the results of qPCR and mRNA-Seq.

**Figure 5 foods-11-02582-f005:**
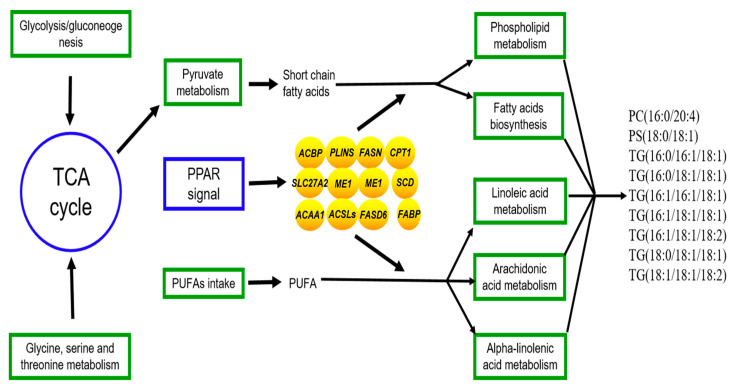
The regulation of genes to lipid metabolism in yak’s LD by the joint analysis of the lipidomics and transcriptome. Oval, box, text without borders represent the crucial DEGs, metabolic pathway or signaling pathway and crucial SDLs. →: regulate or result in.

**Figure 6 foods-11-02582-f006:**
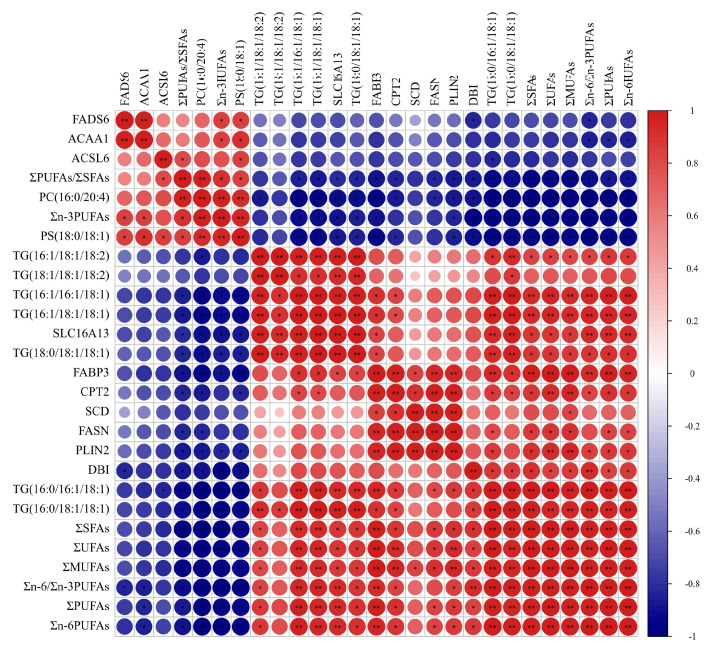
The results of the Pearson correlation analysis for the fatty acids, SDLs and DEGs in yak’s LD. Red and blue circles show the significant positive and negative correlation, respectively. When the color is darker the correlation is higher. * shows *p* < 0.05; ** shows *p* < 0.01.

**Table 1 foods-11-02582-t001:** The composition of the total mixed ration (TMR) and the content of common nutrition and major fatty acid in natural grass and TMR (air-dry basis).

Item	TMR	Natural Grass
Ingredient (%)		
Corn	19.20	-
Wheat bran	9.20	-
Whole corn silage	32.00	-
Oat Hay	28.00	-
Rapeseed meal	8.10	-
NaHCO_3_	1.00	-
NaCl	1.50	-
Premix	1.00	-
Total	100.00	-
Common nutrition (%)		
Crude fat	4.52	2.63
Crude protein	16.96	11.93
Neutral detergent fiber	23.24	76.14
Acid detergent fiber	13.84	10.09
Calcium	0.79	5.22
Phosphorus	0.37	0.07
Fatty acids (%)		
C16:0	0.68	0.30
C18:0	0.21	0.09
C18:1	0.18	0.07
C18:2n6	0.58	0.29
C18:3n3	1.42	0.68

The ingredient of premix was 3000 IU VA, 500 IU VD3, 5 IU VE, 0.1 mg Se, 30 mg Fe, 18 mg Mn, 18 mg Zn and 3 mg Cu in per kg of the diets.

**Table 2 foods-11-02582-t002:** The absolute and relative content of fatty acids in yak longissimus dorsi (LD) muscle under stall feeding (SF) and graze feeding (GF) regimes.

Variable	Absolute Content (mg/100g)			Relative Content (%)		
	GF Yaks (*n* = 6, Mean ± SEM)	SF Yaks (*n* = 6, Mean ± SEM)	F	GF Yaks (*n* = 6, Mean ± SEM)	SF Yaks (*n* = 6, Mean ± SEM)	F
C4:0	2.12 ± 0.08	3.75 ± 0.27	**	0.15 ± 0.01	0.17 ± 0.01	ns
C6:0	0.36 ± 0.02	0.60 ± 0.05	**	0.03 ± 0.001	0.03 ± 0.002	ns
C8:0	0.53 ± 0.02	0.13 ± 0.02	**	0.04 ± 0.001	0.01 ± 0.001	**
C10:0	0.44 ± 0.02	0.61 ± 0.03	**	0.03 ± 0.001	0.03 ± 0.001	ns
C11:0	0.06 ± 0.004	0.09 ± 0.01	**	0.005 ± 0.0002	0.004 ± 0.0003	ns
C12:0	0.37 ± 0.01	0.53 ± 0.05	**	0.03 ± 0.001	0.03 ± 0.002	ns
C13:0	0.79 ± 0.03	1.35 ± 0.17	**	0.06 ± 0.002	0.06 ± 0.01	ns
C14:0	7.67 ± 0.27	10.66 ± 0.36	**	0.56 ± 0.02	0.50 ± 0.02	ns
C14:0iso	0.40 ± 0.05	0.44 ± 0.02	ns	0.03 ± 0.005	0.02 ± 0.001	ns
C15:0	9.00 ± 0.34	7.48 ± 0.65	ns	0.65 ± 0.02	0.35 ± 0.03	**
C15:0iso	1.13 ± 0.06	1.31 ± 0.05	ns	0.08 ± 0.01	0.06 ± 0.003	**
C15:0ai	1.08 ± 0.05	0.88 ± 0.03	*	0.08 ± 0.003	0.04 ± 0.001	**
C16:0	173.53 ± 3.86	335.64 ± 6.60	**	12.61 ± 0.15	15.62 ± 0.36	**
C16:0iso	1.76 ± 0.03	1.68 ± 0.04	ns	0.13 ± 0.004	0.08 ± 0.002	**
C17:0	22.02 ± 1.55	16.79 ± 2.16	ns	1.59 ± 0.09	0.78 ± 0.09	**
C17:0iso	1.74 ± 0.07	1.91 ± 0.06	ns	0.13 ± 0.01	0.09 ± 0.003	**
C17:0ai	2.06 ± 0.08	2.18 ± 0.12	ns	0.15 ± 0.004	0.10 ± 0.01	**
C18:0	326.20 ± 10.16	637.71 ± 9.09	**	23.69 ± 0.39	29.66 ± 0.21	**
C18:0iso	3.81 ± 2.52	1.37 ± 0.08	ns	0.29 ± 0.19	0.06 ± 0.003	ns
C20:0	1.53 ± 0.09	2.17 ± 0.30	ns	0.11 ± 0.01	0.10 ± 0.01	ns
C21:0	4.73 ± 0.28	7.03 ± 0.58	**	0.34 ± 0.02	0.33 ± 0.03	ns
C22:0	2.23 ± 0.22	3.04 ± 0.21	*	0.16 ± 0.02	0.14 ± 0.01	ns
C23:0	9.42 ± 0.31	12.59 ± 2.01	ns	0.68 ± 0.02	0.58 ± 0.09	ns
C24:0	68.21 ± 3.57	94.62 ± 6.14	**	4.95 ± 0.21	4.39 ± 0.24	ns
C14:1	17.45 ± 1.26	1.16 ± 0.13	**	1.27 ± 0.09	0.05 ± 0.01	**
*c-*C15:1	4.88 ± 0.28	3.13 ± 0.32	**	0.35 ± 0.02	0.15 ± 0.02	**
C16:1	35.33 ± 1.52	39.10 ± 4.24	ns	2.57 ± 0.10	1.81 ± 0.18	**
*c-*C17:1	14.55 ± 0.95	6.06 ± 0.81	**	1.06 ± 0.06	0.29 ± 0.04	**
*c*-C18:1	287.11 ± 5.40	459.89 ± 11.29	**	20.87 ± 0.26	21.38 ± 0.32	ns
*t*-C18:1	5.81 ± 0.59	12.18 ± 1.41	**	0.42 ± 0.04	0.57 ± 0.07	ns
*t*11-C18:1	2.99 ± 0.07	3.99 ± 0.09	**	0.22 ± 0.01	0.19 ± 0.004	**
*c-*C20:1	1.71 ± 0.05	5.37 ± 0.51	**	0.12 ± 0.004	0.25 ± 0.03	**
C24:1	37.28 ± 1.06	53.25 ± 3.72	**	2.71 ± 0.09	2.47 ± 0.14	ns
*c*-C18:2n6	166.09 ± 4.86	270.36 ± 5.23	**	12.07 ± 0.25	12.59 ± 0.36	**
*t*-C18:2n6	0.45 ± 0.04	0.58 ± 0.07	ns	0.03 ± 0.003	0.03 ± 0.003	ns
*c*9, *t*11-C18:2	3.32 ± 0.14	3.84 ± 0.14	*	0.24 ± 0.01	0.18 ± 0.01	**
*t*11, *c*15-C18:2	5.14 ± 0.19	6.86 ± 0.24	**	0.37 ± 0.01	0.32 ± 0.01	*
*c-*C18:3n6	10.23 ± 0.30	5.76 ± 0.59	**	0.74 ± 0.02	0.27 ± 0.03	**
C18:3n3	50.31 ± 0.93	33.25 ± 0.72	**	3.66 ± 0.04	1.55 ± 0.04	**
*c*9, *t*11, *t*15-C18:3	0.64 ± 0.02	0.74 ± 0.04	ns	0.05 ± 0.002	0.03 ± 0.002	**
*c-*C20:2	5.94 ± 0.35	3.43 ± 0.60	**	0.43 ± 0.02	0.16 ± 0.03	**
C20:4n6	41.63 ± 1.63	59.06 ± 2.50	**	3.03 ± 0.13	2.74 ± 0.09	ns
*c-*C20:3n3	0.24 ± 0.01	0.27 ± 0.04	ns	0.02 ± 0.001	0.01 ± 0.002	ns
*c-*C20:5n3	41.49 ± 0.93	34.62 ± 1.09	**	3.02 ± 0.09	1.61 ± 0.05	**
*c-*C22:6n3	2.21 ± 0.08	2.71 ± 0.27	ns	0.16 ± 0.01	0.13 ± 0.01	*
ΣSFAs	641.20 ± 16.10	1144.55 ± 14.32	**	46.58 ± 0.45	53.23 ± 0.21	**
ΣMUFAs	407.10 ± 6.21	584.14 ± 14.59	**	29.60 ± 0.19	27.15 ± 0.32	**
ΣPUFAs	327.70 ± 6.88	421.46 ± 4.30	**	23.82 ± 0.37	19.62 ± 0.34	**
ΣUFAs	734.80 ± 11.95	1005.60 ± 14.73	**	53.42 ± 0.45	46.77 ± 0.21	**
Σn-3PUFAs	94.25 ± 1.21	70.86 ± 1.58	**	6.86 ± 0.10	3.30 ± 0.08	**
Σn-6PUFAs	218.41 ± 6.19	335.75 ± 4.62	**	15.88 ± 0.36	15.63 ± 0.32	ns
Σn-6/Σn-3PUFAs	2.32 ± 0.06	4.75 ± 0.15	**	2.32 ± 0.06	4.75 ± 0.15	**
ΣPUFAs/ΣSFAs	0.51 ± 0.01	0.37 ± 0.01	**	0.51 ± 0.01	0.37 ± 0.01	**
ΣBIs	12.09 ± 0.22	15.43 ± 0.30	**	0.88 ± 0.02	0.72 ± 0.01	**
ΣBCFAs	11.98 ± 2.59	9.77 ± 0.18	ns	0.88 ± 0.20	0.45 ± 0.01	ns

*c*: cis; *t*: trans. F: significantly different values as influenced by feeding; *: *p* < 0.05; **: *p* < 0.01; ns: no significant difference. SFAs: saturated fatty acids, UFAs: unsaturated fatty acids, MUFAs: monounsaturated fatty acids, PUFAs: polyunsaturated fatty acids, BCFAs: branched chain fatty acids, BIs: biohydrogenation intermediates. ΣMUFAs: sum of MUFAs, ΣSFAs: sum of SFAs, ΣPUFAs: sum of PUFAs, ΣUFAs: sum of UFAs.

## Data Availability

The original contributions presented in the study are included in the article/Appendix A, and further inquiries can be directed to the corresponding authors.

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
