# Peer review of "Lipidomics and Transcriptome Reveal the Effects of Feeding Systems on Fatty Acids in Yak’s Meat"

_foods, 2022, doi:10.3390/foods11172582_

Round 1

Reviewer 1 Report

The present study seems to be interesting. Authors investigated the differences in fatty acids of yaks fed under two feeding systems (graze feeding (GF) and stall feeding (SF)). Furthermore, in this work, the regulation mechanism of feeding system to fatty acid in the meat of these animals was evaluated. To the best of my knowledge, this article should be published because it provides still unknown information on yak meat (fatty acids phenotype-related gene functions and pathways of fatty acids metabolism) using relatively current techniques such as lipidomics and transcriptome. At the same time, the references employed for its elaboration are generally adequate and it has many recent references. For all the above, I consider that the article should be published in Foods after taking into account the following minor revisions.

Abstract

Line 16: In the abstract, reference must be made to the methods or treatments applied in the elaboration of the study. Briefly describe the main methods or treatments applied if space permits.

Line 17: Consider writing the name of the muscle (longissimus dorsi) in italics, as it is written in Latin. Please note this annotation throughout the manuscript.

Introduction

Line 40: How can meat treat any of the human diseases? Please indicate the appropriate reference here or delete the word "treat" from the sentence.

Line 50: Reference [12] refers to a review article. If possible, try to cite the original source to give credit to the primary authors.

Line 55: Add a space between "consumers" and "[14,15]".

Line 58: Add a space between "industry" and "[16]".

Line 96: Indicate in the text the meaning of the acronym “IMF”, since it is the first time that it is named.

Materials and methods

Line 104: Replace the "," with "and".

Table 1: What ingredient is the "Premix"? Consider indicating this information it in the Table 1 footer. Change the footer of Table 1 to a header. Despite the low percentage (less than 1.5%) the 5 fatty acids indicated in Table 1 are the main fatty acids? "Main” is equal to "majority"?

Line 114: Replace the "," with "and".

Line 117-118: Replace the "and some steps in literature method have been improved" with "some modifications".

Line 123-124: Indicate the instrument employed for the detection of FAMES (also indicate the model and brand as it is done for other instruments, see line 133 as an example).

Line 130: Add a space between "group." and "Total".

Line 138: Add a space between "generated." and "Raw".

Line 143: FPKM or RPKM?

Line 146: Indicate in the text the meaning of the acronym “EP”, since it is the first time that it is named.

Line 169: Add a space between "S1." and "β-actin".

Line 175: Add a space between "ºC." and "Next".

Line 177: Add a space between "Roche." and "Switzerland".

Line 179: Add a space between "PCR." and "Real-time".

Results

Line 216-270: Add a spaces after ":".

Table 2: For clarity, consider adding two additional columns in Table 2, specifying significance (instead of adding asterisks after SEM of SF yaks). I suggest adding the columns after SF yaks in each case (i.e., one column in the absolute content section, and one in the relative content section, as indicate above). The additional columns could be named F (referring to feeding). Thus, "*Significantly different from graze feeding (GF) yaks (P < 0.05); ** Significantly different from GF yak (P < 0.01)" would be replaced in the table footer by "F: significantly different values as influenced by feeding: * (P< 0.05); ** (P< 0.01); ns: no significant difference" or similar.

Example:

Variable

Absolute content

Relative content

GF yaks

SF yaks

F

GF yaks

SF yaks

F

**

**

**

**

ns

ns

…..F: significantly different values as influenced by feeding: * (P < 0.05); ** (P < 0.01); ns: no significant difference……

Line 233: GYs and SYs?

Line 239: Keep the format "up-regulated" or "upregulated" throughout the manuscript.

Line 263: Keep the format "down-regulated" or "downregulated" throughout the manuscript in accordance with the choice regarding up-regulated/ upregulated.

Figure 4: The letters "(a)", "(b)", "(c)", and "(d)" are not displayed correctly in the figure. Please provide a figure where the letters can be seen in their entirety.

Discussion

Line 378: Indicate in the text the meaning of the acronym “TCA”, since it is the first time that it is named.

Line 390: Add a space between "cattle" and "[43,44]".

Line 403: Add a space between "cow" and "[45]".

Line 419: Add a space between "MUFA" and "[50]".

Line 426: Reference [51] refers to a review article. If possible, try to cite the original source to give credit to the primary authors.

Conclussions

Line 441: Replace the "and" with "thus", since the second sentence is a consequence of the first.

Author Response

Response to reviewer comments

Abstract

Line 16: In the abstract, reference must be made to the methods or treatments applied in the elaboration of the study. Briefly describe the main methods or treatments applied if space permits.

Answer: Thank you very much for your comment! The the desrciptions of methods applied in the the study were replenished in the manuscript. “First, the fatty acids in yaks longissimus dorsi (LD) were detected by gas liquid chromatography (GLC).” “Fruther, the transcriptomic and lipidomics profiles in yaks LD were detected by mRNA-Sequencing (mRNA-seq) and ultra-high performance liquid chromatography- mass spectrometry (UHPLC-MS), respectively.” Above changes can be found in line 15-16 and 19-22.

Line 17: Consider writing the name of the muscle (longissimus dorsi) in italics, as it is written in Latin. Please note this annotation throughout the manuscript.

Answer: Thank you very much for your comment! The words “longissimus dorsi” in the manuscript were revised to italics. Such as in line 15,109 and so on.

Introduction

Line 40: How can meat treat any of the human diseases? Please indicate the appropriate reference here or delete the word "treat" from the sentence.

Answer: Thank you very much for your comment! The word "treat" was deleted in the mauscript. You can find the revised description in line 44.

Line 50: Reference [12] refers to a review article. If possible, try to cite the original source to give credit to the primary authors. 

Answer: Thank you very much for your comment! Reference (12) was replaced with a new reference (15). The reference (15) is a original source. This revise is in line 64, 576-577.

Line 55: Add a space between "consumers" and "[14,15]".

Answer: Thank you very much for your comment! A space was added between "consumers" and "[14,15]". This revise is in line 68.

Line 58: Add a space between "industry" and "[16]".

Answer: Thank you very much for your comment! A space was added between between "industry" and "[16]". This revise is in line 72.

Line 96: Indicate in the text the meaning of the acronym “IMF”, since it is the first time that it is named.

Answer: Thank you very much for your comment! The “IMF” is the abbreviation of intramuscular fat. The words “intramuscular fat” were added in line 118-119.

Materials and methods

Line 104: Replace the "," with "and".

Answer: Thank you very much for your comment! The "," was replaced with "and". This revise can be found in line 136.

Table 1: What ingredient is the "Premix"? Consider indicating this information it in the Table 1 footer. Change the footer of Table 1 to a header. Despite the low percentage (less than 1.5%) the 5 fatty acids indicated in Table 1 are the main fatty acids? "Main” is equal to "majority"?

Answer: Thank you very much for your comment! The ingredient of premix was 3000 IU VA, 500 IU VD3, 5 IU VE, 0.1 mg Se, 30 mg Fe, 18 mg Mn, 18 mg Zn and 3 mg Cu in per kg of the diets. The footer of Table 1 was changed to a header. "Main” is equal to "majority". Researches shows that the major fatty acids in grass are C16:0, C18:0, C18:1, C18:2n6 and C18:3n3, which is consistent with the results in this study. In fact, dozens of fatty acis were detected in grass on Tibetan Plateau by us, and the total percentage of C16:0, C18:0, C18:1, C18:2n6 and C18:3n3 is more than 80%. Because above data is only the background information in this study and the article length of this manuscript has been longer at present (more than 9,000 words), only imporatant data was shown in the manuscipt. These revises can be found in line 149-150, 139.

Line 114: Replace the "," with "and".

Answer: Thank you very much for your comment! The "," was replaced with "and" in the revised manuscript. This revise can be found in line 139.

Line 117-118: Replace the "and some steps in literature method have been improved" with "some modifications".

Answer: Thank you very much for your comment! The "and some steps in literature method have been improved" was replaced with "some modifications". This revise can be found in line 153.

Line 123-124: Indicate the instrument employed for the detection of FAMES (also indicate the model and brand as it is done for other instruments, see line 133 as an example).

Answer: Thank you very much for your comment! The residue was redissolved with mobile phase, then was determined on an gas chromatography system (7890A, Agilent Corp., Santa Clara, CA, USA). This revise can be found in line 161-162.

Line 130: Add a space between "group." and "Total".

Answer: Thank you very much for your comment! A space was added between "group." and "Total". This revise can be found in line 169.

Line 138: Add a space between "generated." and "Raw".

Answer: Thank you very much for your comment! A space was added between between "generated." and "Raw". This revise can be found in line 177.

Line 143: FPKM or RPKM?

Answer: Thank you very much for your comment! It is per kilo base of transcripts per million (RPKM). This revise can be found in line 182.

Line 146: Indicate in the text the meaning of the acronym “EP”, since it is the first time that it is named. 

Answer: Thank you very much for your comment! The “EP” is the abbreviation of eppendorf. This revise can be found in line 185.

Line 169: Add a space between "S1." and "β-actin".

Answer: Thank you very much for your comment! The word order here was fully revised in the manuscript. This revise can be found in line 210-211.

Line 175: Add a space between "ºC." and "Next".

Answer: Thank you very much for your comment! A space was added between "ºC." and "Next". This revise can be found in line 217.

Line 177: Add a space between "Roche." and "Switzerland".

Answer: Thank you very much for your comment! A space was added between "Roche." and "Switzerland". This revise can be found in line 219.

Line 179: Add a space between "PCR." and "Real-time".

Answer: Thank you very much for your comment! The description of “The parameter for Real-time PCR” was updated in the manuscipt. This revise can be found in line 221.

Results

Line 216-270: Add a spaces after ":".

Answer: Thank you very much for your comment! A spaces were added after ":" in the mnauscript, such as line 260-263,

Table 2: For clarity, consider adding two additional columns in Table 2, specifying significance (instead of adding asterisks after SEM of SF yaks). I suggest adding the columns after SF yaks in each case (i.e., one column in the absolute content section, and one in the relative content section, as indicate above). The additional columns could be named F (referring to feeding). Thus, "*Significantly different from graze feeding (GF) yaks (P < 0.05); ** Significantly different from GF yak (P < 0.01)" would be replaced in the table footer by "F: significantly different values as influenced by feeding: * (P< 0.05); ** (P< 0.01); ns: no significant difference" or similar.

Answer: Thank you very much for your comment! Your comment is constructive suggestion and opinion. The format of Table 2 was revised according your comments. These revises can be found in line 258-260.

Line 233: GYs and SYs?

Answer: Thank you very much for your comment! "GYs and SYs" was revised to " GF and SF ". This revise can be found in line 279.

Line 239: Keep the format "up-regulated" or "upregulated" throughout the manuscript.

Answer: Thank you very much for your comment! The words "up-regulated" were kept in the whole manuscript, such as in line 344.

Line 263: Keep the format "down-regulated" or "downregulated" throughout the manuscript in accordance with the choice regarding up-regulated/ upregulated.

Answer: Thank you very much for your comment! The words "down-regulated" were kept in the whole manuscript, such as in line 335.

Figure 4: The letters "(a)", "(b)", "(c)", and "(d)" are not displayed correctly in the figure. Please provide a figure where the letters can be seen in their entirety.

Answer: Thank you very much for your comment! The Figure 4 was adjusted, and the letters "(a)", "(b)", "(c)", and "(d)" were displayed correctly now.

Discussion

Line 378: Indicate in the text the meaning of the acronym “TCA”, since it is the first time that it is named.

Answer: Thank you very much for your comment! “TCA” is the abbreviation of tricarboxylic acid. This revise can be found in line 450.

Line 390: Add a space between "cattle" and "[43,44]".

Answer: Thank you very much for your comment! A space was added between "cattle" and "[43,44]". This revise can be found in line 463.

Line 403: Add a space between "cow" and "[45]".

Answer: Thank you very much for your comment! A space was added between "cow" and "[45]". This revise can be found in line 476.

Line 419: Add a space between "MUFA" and "[50]".

Answer: Thank you very much for your comment! A space was added between "MUFA" and "[50]". This revise can be found in line 493.

Line 426: Reference [51] refers to a review article. If possible, try to cite the original source to give credit to the primary authors.

Answer: Thank you very much for your comment! I am very sorry that we don’t find any other relevant literature which can used here.

Conclussions

Line 441: Replace the "and" with "thus", since the second sentence is a consequence of the first.

Answer: Thank you very much for your comment! The "and" was replaced with "thus" in the manuscript. This revise can be found in line 516.

Reviewer 2 Report

The present study deals with the optimization of intramuscular fatty acid profile using animal nutrition, a well-known and researched topic in animal and food science. However, the subject of the study - yak meat - is generally less well-known and researched, which makes this work interesting for the reader. Furthermore, the study clearly benefits from the advanced analytical methods and the multidisciplinary approach involving lipidomics and the expression of selected genes related to fatty acid (FA) metabolism, which may provide new insights for a better understanding of these complex phenomena.

However, as in other members of the bovine family, biohydrogenation in the rumen of yaks plays a crucial role in the metabolism of dietary fats and the resulting FA deposition, and in my opinion, this aspect is not sufficiently considered in the present work. Therefore, this aspect should be elaborated on and discussed in more detail in light of the present results.

Some of the additional information listed below could further improve the text.

Specific comments:

Row:

35: Please remove the space before the comma (and check the rest of the text for similar spelling mistakes).

40: Please add one or more references for this statement.

44-45: Feeding and the differences between ruminants and monogastric animals should also be highlighted.

47: The breeding system is not the best term as it refers more to the selection of animals and mating schemes. I suggest using the term production or feeding system.

48-50: This statement is supported by reference (12), which refers to poultry meat?! As mentioned earlier, there are major differences between ruminants and monogastric in FA metabolism and deposition, and this issue should be elaborated on and considered in much more detail in this paper.

54: Are there any references on the meat composition and FA profile of yak meat?

67-69: Some basic information about FA metabolism in ruminants should be introduced...

70-71: Which genes?

72: Are there any references to yaks (or cattle) here?

74: Please add a reference to the RNA-Seq technique.

78: The concept of the lipidomics approach should be briefly introduced here.

96: To improve the IMF content? IMF content has not been mentioned in the text so far, so this aspect should either be explained in more detail in the introduction or deleted.

100: Were there weight differences between the groups at the beginning of the feeding trial? What were the slaughter weights in the groups SF and GF?

104: When the feed samples are collected and analyzed?

107: Until September? From when? Please indicate the final age of the experimental animals. Also, add some details about the two production systems used (e.g. space per animal)

168: Please give the full name of abbreviations when you mention them for the first time in the text.

186: "at P < 0.05"

Results and Table 2: What was the IMF content of the LD samples? It seems that the SF yaks were fatter at slaughter and probably had a higher IMF content, which could influence the observed differences in the FA profiles of meat.

382-383: What about the phenomena of biohydrogenation in the rumen?

426-427: Not by LA and ALA, but from LA and Ala by the action of desaturases and elongases...

Author Response

Response to reviewer comments

However, as in other members of the bovine family, biohydrogenation in the rumen of yaks plays a crucial role in the metabolism of dietary fats and the resulting FA deposition, and in my opinion, this aspect is not sufficiently considered in the present work. Therefore, this aspect should be elaborated on and discussed in more detail in light of the present results.

Answer: Thank you very much for your comment! We agree with your comments, and your comments provide a constructive guidance to our study. Rumen microorganisms are involved in the transition from UFA to SFA, and this process is commonly referred as the biohydrogenation. Rumen biohydrogenation of UFA can produce CLA and trans-11 oleic acid which is the precursor of CLA synthesis in muscle. Therefore, the deep understanding of the biohydrogenation process of UFA by rumen microorganisms could help to produce ruminant meat being beneficial to human health. The fatty acid metabolism in ruminant is a very complicated process, and is related to rumen microorganisms, liver and adipose tissue. According to your comments, some background knowledges of rumen microorganisms were introduced in the revised manuscript. Such as: “There are significant differences in digestion and absorption characteristics between monogastric and ruminants animals, and the rumen bacteria plays an important part in digestion and absorption of ruminants animals. In production practice, the proportion of concentrate for monogastric animals is usually higher, while the proportion of roughage for ruminants is higher. The rumen microbiome depending on dietary ingredients of diets under different feeding system greatly affect the composition of fatty acids. Due to the hydrogenation of ruminal microorganisms in ruminants, the content of SFAs in ruminants muscle is usually higher than that of monogastric animals and the ratio of PUFA to SFA is lower.” Moreover, the effect of rumen microorganisms to the fatty acids in yaks muscle under different feeding system was discussed in the section of “Discussion”. Because we pay our attention to the fat acids metabolism in intramuscular adipose tissue of yaks in this study and the data of rumen microflora and the fatty acids metabolsim in yak rumen is not obtained by us, this kind of discussion is simple now. Such as: “Fat metabolism in ruminants is closely related to rumen microorganisms and the feeding system directly affects the rumen microorganisms in ruminants. The biohydrogenation by rumen microorganisms in ruminants is the important way of UFAs transferring into MUFAs or SFAs. The differences of rumen microorganisms in yaks under different feeding system can cause the differences of biohydrogenation for UFAs in diet, which is one of major factors that lead to the the differences of fatty acids content in yaks muscle.” These revises are in line 48-56, 431-437. In fact, the study on the effects of ruminal microorganisms to fatty acids metabolism in yaks under different feeding system using 16S rRNA gene sequencing and untargeted metabolomics based on liquid chromatograph-mass spectrometer (LC-MS) is being carried on by us. The resultes will be detailedly shown and discussed in the next manuscript. 

Some of the additional information listed below could further improve the text. Specific comments:

35: Please remove the space before the comma (and check the rest of the text for similar spelling mistakes).

Answer: Thank you very much for your comment! The space was removed. The rest of the text for similar spelling mistakes was checked too. This revise can be found in line 38.

40: Please add one or more references for this statement.

Answer: Thank you very much for your comment! The references (8) for this statement was added in the manuscript. This revise can be found in line 45, 561-562.

44-45: Feeding and the differences between ruminants and monogastric animals should also be highlighted.

Answer: Thank you very much for your comment! The descriptions of feeding and the differences between ruminants and monogastric animals were replenished in the manuscript. There are significant differences in digestion and absorption characteristics between monogastric and ruminants animals, and the rumen bacteria plays an important part in digestion and absorption of ruminants animals. In production practice, the proportion of concentrate for monogastric animals is usually higher, while the proportion of roughage for ruminants is higher. The rumen microbiome depending on dietary ingredients of diets under different feeding system greatly affect the composition of fatty acids. Due to the hydrogenation of ruminal microorganisms in ruminants, the content of SFAs in ruminants muscle is usually higher than that of monogastric animals and the ratio of PUFA to SFA is lower [9,10]. These revises can be found in line 48-56.

47: The breeding system is not the best term as it refers more to the selection of animals and mating schemes. I suggest using the term production or feeding system.

Answer: Thank you very much for your comment! According to your suggestion, the words “feeding system” was used here in revised manuscript. This revise can be found in line 61.

48-50: This statement is supported by reference (12), which refers to poultry meat?! As mentioned earlier, there are major differences between ruminants and monogastric in FA metabolism and deposition, and this issue should be elaborated on and considered in much more detail in this paper.

Answer: Thank you very much for your comment! Because of our carelessness, the reference (12) was wrongly cited in the manuscript. The new reference (15) was used in the revised manuscript, and is the study on meat quality and fatty acids of lambs under different feeding system. This revise can be found in line 64, 576-577.

54: Are there any references on the meat composition and FA profile of yak meat?

Answer: Thank you very much for your comment! The references (19), (20) and (21) on the meat composition and FA profile of yak meat were replenished in revised manuscript. The content of SFAs in yaks muscle was lower than the values of cattle, while the contents of PUFAs and monounsaturated fatty acids (MUFAs) in yaks muscle were higher than the values of cattle [19,20]. The ratio of n-6 PUFAs/ n-3 PUFAs in muscle of yak calf was 1.15 [21]. These revises can be found in line 68-71, 585-590.

67-69: Some basic information about FA metabolism in ruminants should be introduced...

Answer: Thank you very much for your comment! Some basic information about FA metabolism in ruminants was replenished in the revised manuscript. Compared with monogastric animals, the fat metabolism in ruminants is more complex. The rumen microorganisms in ruminants can translate fatty acids from forage or feed into new fatty acids [26], and finally these new fatty acids deposit in subcutaneous, muscle and abdomen. These revises can be found in line 84-87, 601-603.

70-71: Which genes?

Answer: Thank you very much for your comment! These genes include LIPC, ERFE, FABP3, PLA2R1, LDLR, and SLC10A6 and so on. Above genes were replenished in the revised manuscript. These revises can be found in line 91-92.

72: Are there any references to yaks (or cattle) here?

Answer: Thank you very much for your comment! The reference (32) on the study of fatty acids in yaks with transcriptomics was added in the revised manuscript. The reference on the study of fatty acids in yaks with lipidomics was not found by us. This revise can be found in line 94, 617-619.

74: Please add a reference to the RNA-Seq technique.

Answer: Thank you very much for your comment! This sentence expresses the goal of transcriptomic analysis for yak muscle under different feeding system in this study, so reference is not needed here. Moreover, the studies for the yaks are less, and there is no report on the transcriptomic analysis for yak muscle under different feeding system at present.

78: The concept of the lipidomics approach should be briefly introduced here.

Answer: Thank you very much for your comment! Lipidomics is a kind of high-throughput analysis technology, and can systematically analyze the changes of lipid composition and expression in organisms [33] and explore the functions of lipid families and lipid molecules in various biological processes [34]. Above content was replenisehed in revised manuscript in line 96-99.

96: To improve the IMF content? IMF content has not been mentioned in the text so far, so this aspect should either be explained in more detail in the introduction or deleted.

Answer: Thank you very much for your comment! “IMF” is the abbreviation of intramuscular fat, and the complete spelling was replenished in line 118-119.

100: Were there weight differences between the groups at the beginning of the feeding trial? What were the slaughter weights in the groups SF and GF?

Answer: Thank you very much for your comment! In the beginning, the weight of yaks were 210.33 ± 10.23 kg in line 125 and all yals were randomly divided into SF and GF group, so there were no difference between the two groups at the beginning of the feeding trial. The slaughter weights of yaks in SF and GF groups were 304.88 ± 10.62 and 382.55 ± 8.67 kg, respectively, and there were significant difference in yaks weight between SF and GF.

104: When the feed samples are collected and analyzed?

Answer: Thank you very much for your comment! The grass sample was collected on September. This revise can be found in line 136-137.

107: Until September? From when? Please indicate the final age of the experimental animals. Also, add some details about the two production systems used (e.g. space per animal)

Answer: Thank you very much for your comment! Two year old yaks were selected on May and the feeding test last five mounths, so the final age of yaks is bout two years and five months. The GF yaks were only grazed in natural pasture with no supplements, and the SF yaks were fed with total mixed ration (TMR) in stall. These yaks in the same group can freely eat grass or MTR and drink water by themselves, respectively. All experimental animals were dewormed before the test. Above information are in line 122-134.

168: Please give the full name of abbreviations when you mention them for the first time in the text.

Answer: Thank you very much for your comment! “EP” is the abbreviation of eppendorf. This revise can be found in line 185.

186: "at P < 0.05"

Answer: Thank you very much for your comment! The word was revised in the manuscript, and this revise can be fiund in line 229.

Results and Table 2: What was the IMF content of the LD samples? It seems that the SF yaks were fatter at slaughter and probably had a higher IMF content, which could influence the observed differences in the FA profiles of meat.

Answer: Thank you very much for your comment! The SF yaks were indeed fatter at slaughter and had a higher IMF content. The IMF content in yaks muscle under GF and SF were 1.74 and 2.76% on September, respectively. The fatty acids are one of important components for IMF, and the fatty acids content decide the IMF content in yaks. Therefore, IMF dont’t influence the fatty acids in yaks muscle, but the fatty acids influence the IMF.

382-383: What about the phenomena of biohydrogenation in the rumen?

Answer: Thank you very much for your comment! Fat metabolism in ruminants is closely related to rumen microorganisms and the feeding system directly affects the rumen microorganisms in ruminants. The biohydrogenation by rumen microorganisms in ruminants is the important way of UFAs transferring into MUFAs or SFAs [50]. The differences of rumen microorganisms in yaks under different feeding system can cause the differences of biohydrogenation for UFAs in diet, which is one of major factors that lead to the the differences of fatty acids content in yaks muscle. These revises can be found in line 431-437.

426-427: Not by LA and ALA, but from LA and Ala by the action of desaturases and elongases...

Answer: Thank you very much for your comment! The sentence was revised in the manuscript. The n-6 PUFAs like arachidonic acid are derived from linoleic acid by the action of desaturases and elongases, and n-3 PUFAs like DHA and EPA are derived from linolenic acid. These revises can be found in line 500-502.